# Opposing patterns of abnormal D1 and D2 receptor dependent cortico-striatal plasticity explain increased risk taking in patients with DYT1 dystonia

Tom Gilbertson[1,2]*, David Arkadir[3], J. Douglas Steele[1,2]

**1** Department of Neurology, Ninewells Hospital & Medical School, Dundee, United Kingdom, **2** Division of Imaging Science and Technology, Medical School, University of Dundee, Dundee, United Kingdom, **3** Department of Neurology, Hadassah Medical Center and the Hebrew University, Jerusalem, Israel

* tgilbertson@dundee.ac.uk

**Data Availability Statement:** All relevant data are within the manuscript and its Supporting Information files.

## Abstract

Patients with DYT1 dystonia caused by the mutated TOR1A gene exhibit risk neutral behaviour compared to controls who are risk averse in the same reinforcement learning task. It is unclear whether this behaviour can be linked to changes in cortico-striatal plasticity demonstrated in animal models which share the same TOR1A mutation. We hypothesised that we could reproduce the experimental risk taking behaviour using a model of the basal ganglia under conditions where cortico-striatal plasticity was abnormal. As dopamine exerts opposing effects on cortico-striatal plasticity via different receptors expressed on medium spiny neurons (MSN) of the direct (D1R dominant, dMSNs) and indirect (D2R dominant, iMSNs) pathways, we tested whether abnormalities in cortico-striatal plasticity in one or both of these pathways could explain the patient's behaviour. Our model could generate simulated behaviour indistinguishable from patients when cortico-striatal plasticity was abnormal in both dMSNs and iMSNs in opposite directions. The risk neutral behaviour of the patients was replicated when increased cortico-striatal long term potentiation in dMSN's was in combination with increased long term depression in iMSN's. This result is consistent with previous observations in rodent models of increased cortico-striatal plasticity at in dMSNs, but contrasts with the pattern reported *in vitro* of dopamine D2 receptor dependant increases in cortico-striatal LTP and loss of LTD at iMSNs. These results suggest that additional factors in patients who manifest motor symptoms may lead to divergent effects on D2 receptor dependant cortico-striatal plasticity that are not apparent in rodent models of this disease.

## Introduction

Cortico-striatal plasticity has been implicated in the acquisition and extinction of learned actions through positive [1] and negative reinforcement learning [2]. Optogenetic studies have confirmed a causal role for phasic dopamine in the form of the reward prediction error signal

**Funding:** The work was supported by a grant to TG from Dystonia UK and the University of Dundee movement disorders endowment fund.

**Competing interests:** The authors have declared that no competing interests exist

in determining behavioural choices [3, 4]. This has led to the widely accepted view that dopamine modifies behaviour by mediating its opposing effects on cortico-striatal synaptic strength via the two principle subtypes of dopamine receptor [5]. Within this framework, phasic increase in dopamine, which accompanies a rewarding outcome, strengthens the cortico-striatal synapse at Medium Spiny Neurons within the "direct" or striato-nigral pathway (dMSNs). As this pathway exerts a net facilitatory influence on thalamo-cortical excitability, cortico-striatal synaptic potentiation in the direct pathway increases the likelihood of this choice being repeated [6, 7]. Conversely, phasic decreases in dopamine associated with an aversive outcome lead to a strengthening of cortico-striatal synapses within the "indirect" or striato-pallidal pathway (iMSNs). As the indirect pathway exerts an inhibitory influence on thalamo-cortical excitability, the effect of increased cortico-striatal synaptic strengthening at iMSNs is to suppress the likelihood of a choice with an aversive outcome being repeated [6, 7]. Both of these signals rely upon the induction of cortico-striatal long-term potentiation (LTP) at MSNs to mediate their behavioural effect, albeit under opposite phasic changes in dopamine [8, 9]. The effects of dopamine on plasticity at the cortico-striatal synapse are in turn mediated by the predominant expression of D1 (D1R) and D2 (D2R) receptors on MSNs of the direct (dMSN) and indirect (iMSN) pathways respectively [5, 10]. Accordingly, in humans, individual sensitivity to positive and negative feedback correlates with the extent of D1 or D2 receptor expression and genetic influences on their variability [11–13].

The mutated *TOR1A* gene causes generalised dystonia (DYT1), a movement disorder characterised by sustained or intermittent muscle contractions leading to abnormal repetitive movements and postures [14]. Brain slice recordings of MSNs from rodents expressing the human mutant gene exhibit abnormal cortico-striatal plasticity with a combination of abnormally strong long term potentiation, LTP [15] and weak long-term depression, LTD [15, 16]. Subsequent studies have delineated a receptor specific abnormality in D2R expression as the principle cause for impaired LTD at the cortico-striatal synapse in this model [17, 18]. In view of the importance of cortico-striatal plasticity in reinforcement learning, Arkadir et. al., proposed that patients with the TOR1A mutation should exhibit a learning strategy that is contingent with the abnormal plasticity seen in rodent models [15, 19, 20]. The patients in this study were found to be significantly more likely to make a risky choice in a reinforcement learning task compared to controls. They concluded that this risk taking behaviour was consistent with asymmetric integration of the phasic dopamine signal as a consequence of maladaptive striatal plasticity. Given the distinct effects that these signals mediate on direct and indirect pathway excitability, they proposed three possible abnormalities of cortico-striatal plasticity at dMSNs (D1R dominant) and iMSNs (D2R dominant) that may lead to the pattern of observed behaviour: 1) An increased sensitivity to a "win," due to increased LTP at dMSNs with intact iMSN plasticity, 2) increased sensitivity to a "win," with blunted sensitivity to a "loss" both due to abnormally increased cortico-striatal LTP in dMSNs and increased LTD at iMSN cortico-striatal synapses, 3) increased LTP at cortico-striatal synapses in dMSNs and iMSNs with simultaneously blunted LTD at in both types of MSNs. The third explanation was favoured as it was consistent with the pattern observed from the rodent slice data. This pattern is nevertheless the most difficult of the three to reconcile with increased risk taking behaviour. If it were the underlying cause, any increased riskiness mediated by pathological LTP at dMSNs would be acting in opposition to the risk aversive effects of increased LTP on the excitability of iMSNs. In this scenario, increased risk taking could therefore only be conferred by an abnormality in cortico-striatal plasticity in the dMSN population that was substantially greater than that in the indirect pathway iMSN population.

We wanted to address this conflict between the reported plasticity abnormalities demonstrated in rodent models and the risk taking behaviour observed in patients using a model of

cortico-striatal plasticity [21]. In these simulations the model reproduced decision making in the task whilst being forced to learn under the three proposed conditions of abnormal striatal plasticity. We found the combination of cortico-striatal plasticity abnormality in dMSN and iMSN reported from the rodent experiments was least robust for reproducing the actual experimental behaviour of patients. In contrast, the model generated simulated behaviour that was statistically indistinguishable from that observed experimentally by patients, only when learning under conditions with the opposite pattern of iMSN cortico-striatal plasticity abnormality (reduced LTP / increased LTD). We propose this abnormality is easily reconciled with current understanding of the neurobiology of learning and increased risk taking. Notably, we suggest that D2R dysfunction may fundamentally differ between dystonically manifest patients and non-dystonically manifest animal models which share the *TOR1A* gene mutation.

## Methods

### Subjects and behavioural paradigm

Behavioural data was from Arkadir et. al., (2016) [19] which included 13 adult patients with DYT1 dystonia and 13 age and sex-matched controls. All participants gave written informed consent and the study was approved by the Institutional Review Boards of Columbia University, Beth Israel Medical Center, and Princeton University. Further details regarding patient medications and clinical assessments are described in detail in the original manuscript. The trial-and-error (reinforcement) learning task consisted of 326 trial presentations of four pseudo-letters which served as cues ('slot machines'). This included an initial familiarisation (training phase) of 26 trials. Each cue was associated with a different reward schedule (sure 0¢, sure 5¢, sure 10¢, with the so-called '"risky" cue associated with 50:50% probabilities of 0¢ or 10¢ payoffs). The task consisted of pseudo-randomised presentations of the cues in either "forced" or "choice" trials (Fig 1). Pay-out feedback was presented following a "forced" trial when one of the four cues was presented on its own and selected. During a "choice" trial, feedback was given following the subject's choice of one cue from the pair presented. One of five pairs of cue combinations were presented during "choice" trials. These included 0¢ versus 5¢, 5¢ versus 10¢, 0¢ versus 0/10¢, 5¢ versus 0/10¢ and 10¢ versus 0/10¢. The principle behavioural result reported by Arkadir et. al., was an increased tendency for patients to choose the risky cue when presented with the 5¢ versus 0/10¢ pairing. We therefore focused our re-analysis of their data on these "risk" choice trials highlighted by Arkadir et. al. To ensure consistency with their analysis of the task behaviour, we report in an identical fashion, the overall proportion of risky cue choice both across the task (n = 60 trials) as a whole (Fig 1A) and across four (n = 15 trial) blocks (Fig 1B).

### Model fitting

The behavioural data was fitted to the cortico-striatal plasticity (CSP) model described in detail in Gilbertson et. al. (2019) [21]. This combines a traditional temporal difference (TD) model of reinforcement learning with biologically plausible cortico-striatal synaptic weight changes based on *in vitro* data [5]. At the core of this model are two striatal MSN populations, representing the direct (dMSN) and indirect (iMSN) pathways, which differ in their dominant expression of D1R (direct pathway) and D2R (indirect pathway) dopamine receptors. The outputs of these pathways are in turn a function of the interaction between the reward prediction error (RPE) $(R(t) - Q(A, t - 1))$ signal in the equation;

$$Q(A, t) = Q(A, t - 1) + \alpha(R(t) - Q(A, t - 1)) \tag{1}$$

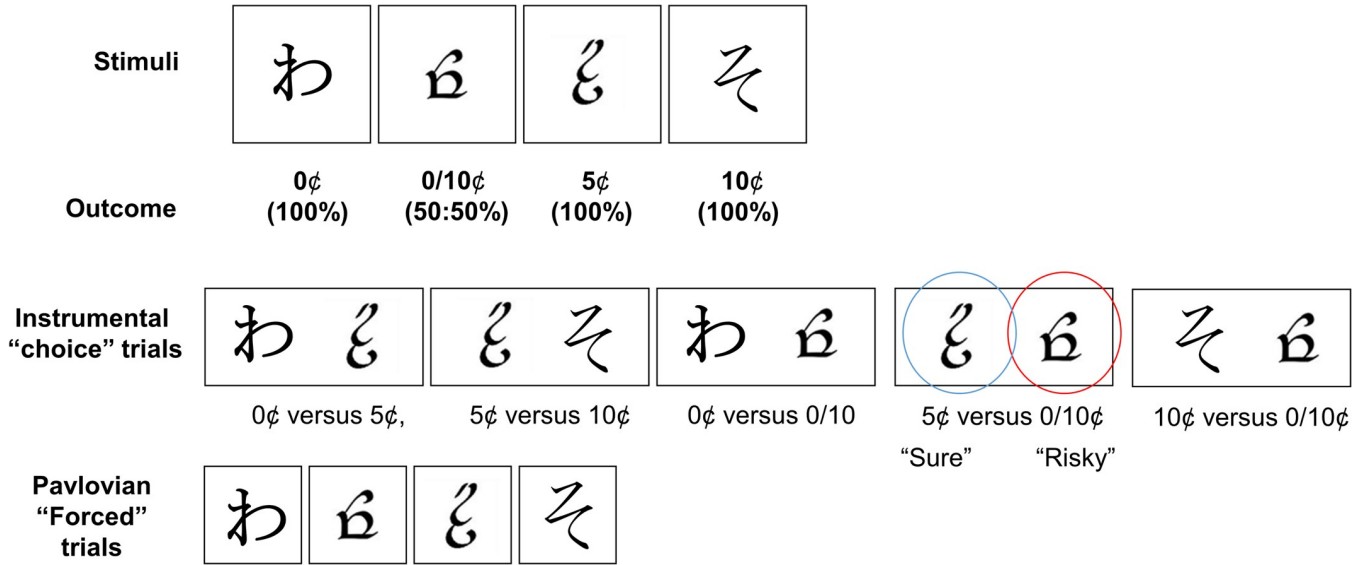

**Fig 1. Task.** Examples of visual stimuli used in the reinforcement learning task by Arkadir et. al., 2016. Trials were randomly presented as either single stimuli which required a forced choice and corresponding outcome or as instrumental trials where subjects were instructed to choose one of two of the stimuli. The risky cue choice trials were between the "risky cue" whose choice led to a 50% chance of 10¢ or 0¢ (highlighted here by the red circle) or the "sure cue" which had a 100% chance of 5¢ payout.

where $\alpha$ is the learning rate, $R_t$ is the outcome (reward[1] or nothing[0]), and the striatal activity $S_n$ of each population on trial $t$ for action $A$, which was defined as:

$$S_n(A, t) = W_n(A, t-1) \cdot c \tag{2}$$

where $W$ is the cortico-striatal synaptic weight and c is a constant input of 1. Each population $S_{dMSN}$(dMSNs) and $S_{iMNS}$ (iMSNs) represented four actions (corresponding to the four cue choices in the task). The cortico-striatal synaptic weights in each population are modified at the synapse corresponding to the chosen action $A$;

$$W_n(A, t) = \begin{cases} W_n(A, t-1) + \Delta W_n(A, t-1), & \text{if} \quad W_n(A, t-1) + \Delta W_n(A, t-1) > 0 \\ 0, & otherwise \end{cases} \tag{3}$$

With the change in synaptic weight being the product of the striatal postsynaptic activity and the influence of dopamine:

$$\Delta W_n(A, t) = \Delta d_n(t) \cdot S_n(A, t). \tag{4}$$

Here the magnitude $\Delta d_n$ of dopamine's effect on synaptic plasticity is:

$$\Delta d_n(t) = \begin{cases} a_n(DA(t) - \theta), \text{if } DA(t) > \theta \\ b_n(DA(t) - \theta), \text{otherwise} \end{cases} \tag{5}$$

where $(a_n, b_n)$ are coefficients determining the dependence of synaptic plasticity on the current trial's level of dopamine DA(t), and the constant $\theta$ determines the baseline level of dopamine.

Eq 5 links the RPE from Eq 1 by;

$$DA(t) = DA_{min} + \frac{(RPE(t) - RPE_{min})DA_{range}}{RPE_{range}} \tag{6}$$

where RPE(t) < 0, $DA_{min} = 0$, $DA_{range} = \theta$, $RPE_{min} = -1$, $RPE_{range} = 1$; otherwise $DA_{min} = \theta$, $DA_{range} = 1 - \theta$, $RPE_{min} = 0$, $RPE_{range} = 1$.

For forced trials the striatal population's weight $W_n(A,t)$ is updated for the forced action choice only. During choice trials the model's chosen action is determined by competition between the two striatal pathways for control of the pallidal output: see Bariselli et. al., 2018 [22] for a review of the evidence for competition between direct and indirect pathways. The striatal weights were then updated for the action chosen from the pair of choices. Thus, for a choice trial with two actions $(A_1,A_2)$;

$$GPi(A_1, t) = (S_{dMSN}(A_1, t) - S_{iMSN}(A_1, t))H(S_{dMSN}(A_1, t) - S_{iMSN}(A_1, t)). \tag{7}$$

where $H()$ is the Heaviside step function: $H(x) = 0$ if $x \leq 0$, and $H(x) = 1$ otherwise; and similarly for action $A_2$. In turn the probability of choosing action $A_1$ was determined by the softmax equation with the basal ganglia's output substituted for the value term:

$$P(A_1, t) = \frac{e^{(GPi(A_1,t)/\beta)}}{e^{(GPi(A_1,t)/\beta)} + e^{(GPi(A_2,t)/\beta)}}. \tag{8}$$

The CSP model requires estimation of six free parameters. This includes two relating to the phasic dopamine (RPE) signal, namely the learning rate ($\alpha$) and reward sensitivity or inverse temperature parameter ($\beta$), and four parameters which govern the magnitude of cortico-striatal plasticity at dMSNs: $a_1$ (LTD) $b_1$ (LTP); and at iMSNs: $a_2$ (LTP), $b_2$ (LTD). Each of these parameters govern the gradient of the synaptic weight change function and its interaction with phasic dopamine. Larger values of each parameter lead to more significant changes in synaptic weight across the dynamic range of dopamine, as this is encoded in the phasic increases and decreases that index the choice outcomes.

Estimation of the 6 parameters ($a_1$, $b_1$, $a_2$, $b_2$, $\alpha$, $\beta$) was performed simultaneously using data from the whole task including all trials of both types (forced and choice) and the initial training phase. We optimised the model parameters by minimising the negative log likelihood of the data given each parameter combination. This was done using the Matlab (Mathworks, NA) function *fmincon*. The initial starting points of this function were estimated following a grid-search of the parameter space. The bounds of both *fmincon* and the grid-search were defined as $a_1 = [0, 2.5]$; $b_1 = [0, 1.5]$; $a_2 = [-2.5,0]$, $b_2 = [-1.5,0]$, $\alpha = [0, 1]$, $\beta = [0, 2]$. (The softmax equation in the CSP model divides by $\beta$ hence the range here has low values relative to TD models where $\beta$ multiplies). The intervals for the grid-search were 0.2, for the "$a$" parameters, 0.1 for the "$b$" parameters and 0.1 and 0.2 for $\alpha$ and $\beta$ respectively due to allow for the differences in the ranges of their bounds.

Probability density functions for each of the four plasticity parameters were generated by fitting a nonparametric kernel function to control subject's estimates. These were used to determine the parameter space bounds that defined "pathologically" high (>95%) or low (<5%) plasticity within the model's parameter space. For hypotheses testing where cortico-striatal plasticity was considered to be within the normal "physiological" range, the bounds were defined by the 5% and 95% confidence limits of the control subject values. Fitting was then performed separately for each hypothesis (**H1**-**H3**) in turn. The combination of dMSN and iMSN cortico-striatal plasticity abnormalities for each hypothesis were:- "**H1**" : Increased dMSN LTP & decreased dMSN LTD; "**H2**" : Increased dMSN LTP & decreased dMSN LTD,

Decreased iMSN LTP & increased iMSN LTD; "**H3**" : Increased dMSN LTP & decreased dMSN LTD, Increased iMSN LTP & decreased iMSN LTD.

## Results

### Controls

To test the reliability of the final model fitting and its ability to capture healthy control behaviour, experimental data sets (n = 1000) were simulated, using the final parameter estimates (See Table 1 for values). These simulations were generated using the final individual subject parameters incorporated into the CSP model re-performing the task with the original experimental cue sequence. We compared the simulated model decisions to choose the risky cue to the choice probabilities from the control subject's experimental data, by performing a two-way ANOVA with two independent variables: source of choices [e.g. simulation, experiment], block number [1–4]. There was no significant difference in the probability of choosing the risky cue in the experimental behavioural data or the simulated behavioural data (ANOVA, F (1) = 0.01, p = 0.91), or any difference between the simulated or experimental risky choices across the four blocks of the tasks (ANOVA [Source, Block], F (3) = 0.4. p = 0.75). For an illustrative comparison, the experimental probabilities of choosing the risky cue are plotted in blue for the controls in Fig 2, with both experimental and simulated choices overlaid in Fig 3. This analysis suggests that the average choice behaviour between each block in the task could be simulated using the CSP model for individual controls, and that this was statistically indistinguishable from that seen experimentally.

### Patients

Re-analysing the experimental data of Arkadir et al., we found the same tendency for patients to show significantly less risk aversion (Fig 2A), although choosing the risky stimulus significantly more often than controls (DYT 0.44 ± 0.04, CTL 0.26 ± 0.05, Mann-Whitney z = 2.23, df = 24, P < 0.05). Importantly, the patients increased risky decision taking continued throughout the four experimental blocks (conducting a one-way ANOVA with task block as a single independent variable, indicating no significant effect of block, F (1) = 0.62, p = 0.61). Notably the choice of the risky cue led to a 50:50% probability of either 0 or 10$ outcome, so this absence of any modification of risk taking behaviour over time was despite receiving proportionately more 0$ (losing) outcomes (Fig 2B). Our aim of fitting the patient's behaviour data was therefore to capture both the overall level of riskiness across the task and this absence of risky cue devaluation between blocks. We therefore re-fitted the patient's behavioural data whilst constraining the bounds of the fitting procedure to the parameter space defined by the three hypothesised plasticity combinations (H1-H3). As all three hypothesis shared an increase in dMSN cortico-striatal LTP in common, each individual hypothesis was aimed at testing different contributions of iMSN cortico-striatal plasticity for risk taking. For "H1" Increased dMSN LTP was accompanied by physiological (control) levels of iMSN cortico-striatal

**Table 1. Final model parameter estimates.**

| | Cortico-striatal plasticity parameters | | | | Temporal difference parameters | |
|---|---|---|---|---|---|---|
| | dMSN LTD (a1) | dMSN LTP (b1) | iMSN LTP (a1) | iMSN LTD (b2) | $\alpha$ | $\beta$ |
| Patients (H1) | 0.21±0.01 | 1.45±0.01 | -1.69±0.16 | -0.92±0.15 | 0.72±0.07 | 0.09 0.06 |
| Patients (H2) | 0.19±0.02 | 1.47±0.03 | -0.38±0.015 | -1.55±0.023 | 0.53±0.09 | 0.14±0.03 |
| Patients (H3) | 0.15±0.017 | 1.45±0.01 | -2.44±0.01 | -0.31±0.02 | 0.80±0.07 | 0.10±0.01 |
| Controls | 1.42±0.24 | 1.02±0.09 | -1.44 ±0.22 | -1.07±0.12 | 0.34 ±0.1 | 0.09 ±0.01 |

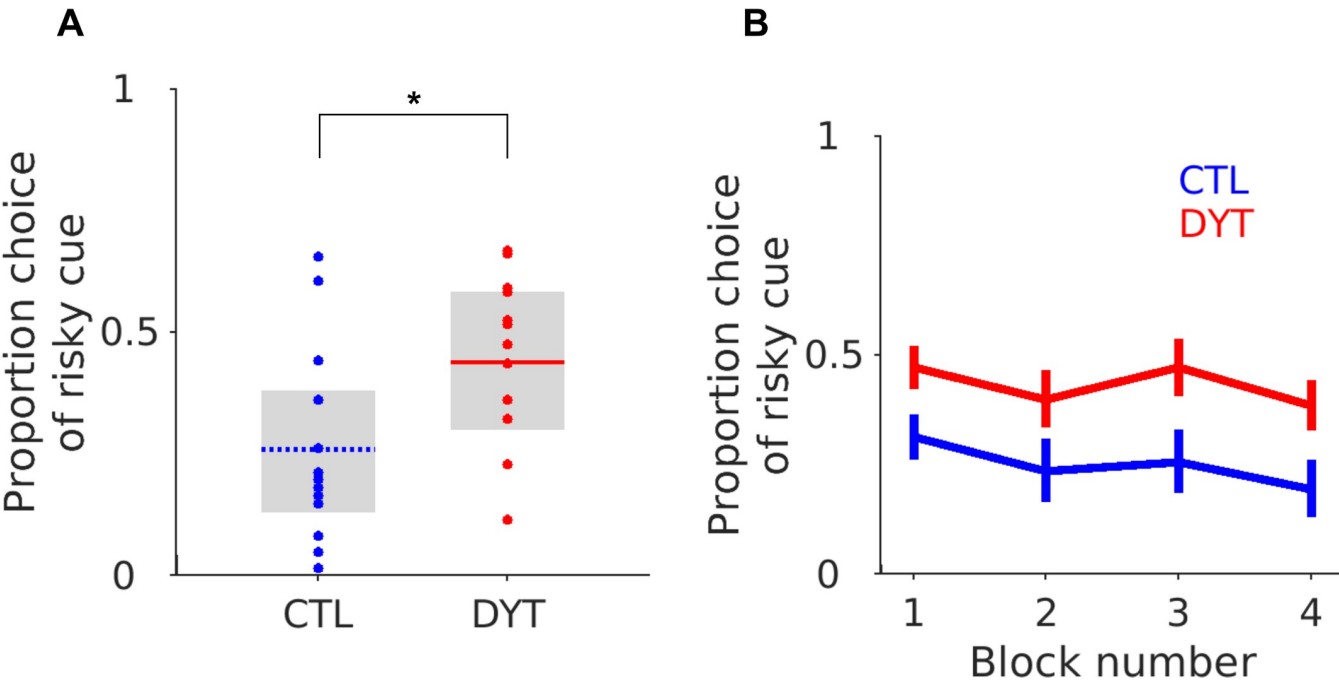

**Fig 2. Experimental risk taking behaviour in patients and controls.** (**A**) Boxplots illustrate the mean choice probability of the patients (red) and controls (blue) represented by the horizontal lines across the task as a whole. Each individual subjects choice probabilities are superimposed. The grey boxes represent the interquartile range. (**B**) The patients and controls average choice probabilities across four 15 trial blocks over the course of the task. The error bars represent the S.E.M. * Mann-Whitney z = 2.33, P < 0.05.

plasticity. For "H2" iMSN cortico-striatal plasticity was opposite to that for dMSNs and baised towards excess LTD. Finally, for "H3" the increase in dMSN LTP was accompanied by a parallel increase in LTP at iMSN cortico-striatal synapses. Comparing the individual negative log likelihoods of each hypothesis demonstrated a trend towards H1 and H2 (10 subjects) explaining the behaviour better than H3 (Fisher exact test $\chi^2$ (24) = 11.1, p = 0.05 Bonferroni corrected), but no overall single wining hypothesis. Given the similarity of both the negative log likelihood values and the overlap between the hypothetical plasticity abnormalities, we tested whether any one of the hypothesis could recover the risky choice behaviour by comparing their simulated (generated) risk taking behaviour. We generated simulated "experiments" (n = 1000) using individual patient parameter estimates for each hypothesis. The results are plotted alongside the simulated and experimental control data in Fig 3. As illustrated (Fig 3B) the only hypothesis, which could accurately recover the experimental behaviour, was H2 with LTP increased at dMSNs and LTD increased at iMSN cortico-striatal synapses. A feature of the alternative hypotheses (H1 & H3) was their inability to capture the between-block risk taking behaviour of the patients which remained relatively similar across the whole task (i.e. from blocks 1–4 the risky cue was chosen to a similar degree). In contrast, when the model performed the task with the predefined plasticity abnormalities associated with H1 & H3, the models choice probability of the risky cue substantially reduced between the beginning (block 1) and end of the task (block 4).

Statistically, this observation was reflected by there being no discernible difference between the simulated models risky cue choice, under H2's plasticity conditions, and the experimental patient's risky cue choice probability. A two-way ANOVA with two independent variables (source of choices [simulation or experiment], task block) indicated there was no effect of the source of the choice data (ANOVA, F (1) = 0.44, p = 0.50) or any significant interaction

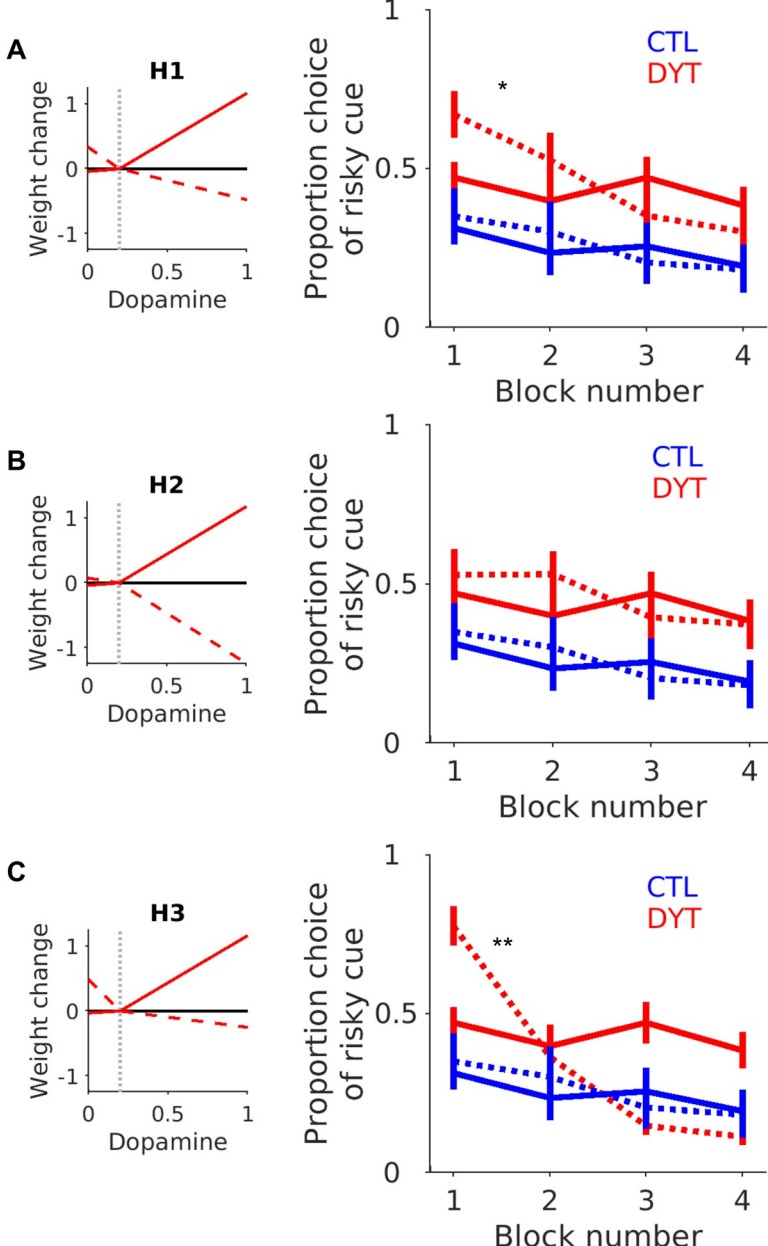

**Fig 3. Simulated risk taking under each hypothetical plasticity abnormality.** Each plot from **A**-**C** illustrates the final average synaptic weight change curve for the patients under each hypothetical plasticity condition (H1-H3). See text for details of the dMSN and iMSN plasticity abnormality for each hypothesis. The average simulated (n = 1000 simulations) choice probability of the risky cue for each block (1–4) in the task is represented by the dashed (—) lines with the patients in red and controls in blue. The error bars represent the average standard error across the simulated experiments. The solid lines (-) represent the average choice (±S.E.M) from the experimental data of Arkadir et. al. (2016). Significant differences between the simulated and experimental mean choice probability were present under plasticity conditions for H1 (*p = 0.01) and H3 (**p<0.001) but not for H2, consistent with the overlapping experimental and simulated choices for this hypothesis.

between the variables (ANOVA, F(3) = 1.48, p = 0.21). Consistent with the experimental choice behaviour in the patients, there was no statistically significant between-block differences in choice probability for the simulations under H2's plasticity conditions (ANOVA, F(3)

= 1.99, p = 0.12). In contrast, there was a significant difference in the simulated decision making of the model under the plasticity conditions of H1 and H3. For both hypotheses there was a significant interaction between the variables for H1 (ANOVA, F (3) = 3.63, p = 0.01) and H3, (ANOVA, F (3) = 32.12, p<0.001). Furthermore, there was also an effect of block for both hypotheses, H1, (ANOVA, F(3) = 5.46, p<0.01), H3 (ANOVA, F(3) = 43.49, p<0.001). The choice probability across the task for both of these models therefore contrasted with, and did not capture, the experimental patient behaviour where no statistical difference was detected between each block of the task (see above). In all, this analysis would support the assertion that the only hypothesis that could accurately reproduce both the risk neutral behaviour of the patients and their behaviour between blocks across the task, was one where LTP was increased in dMSNs in combination with increased LTD at iMSN cortico-striatal synapses. The reliability of the model under the plasticity conditions of H2 to replicate the experimental behaviour is further illustrated in Fig 4A. Here we plot a single simulated experiment and for illustrative proposes, a random sample of 100 (from the 1000 generated) simulated control and dystonia behavioural experiments.

Consistent with the constraints on the fitting procedure for H2, where all four parameters were in the "pathological" range, the final plasticity parameters fitted to the patients ($a_1$-$b_2$) were all significantly different from the healthy controls (Two-way ANOVA (F(3) = 30, p<0.001). In contrast, there was no corresponding difference in the $\alpha$ (Mann-Whitney z (24) = 1.2, p = 0.23), or $\beta$ terms (Mann-Whitney z (24) = 1.2, p = 0.22). The final dopamine weight change curve for patients (H2) and controls illustrates the expected effects of dopamine in the presence of increased D1R mediated dMSN LTP to LTD and decreased D2R mediated LTP to LTD at iMSN cortico-striatal synapses (Fig 5). Relative to controls, patients significantly strengthened the direct pathway (dMSN's) and weakened the cortico-striatal synaptic connection in the indirect pathway (iMSN) in response to a phasic increase in dopamine. Conversely, when dopamine levels are

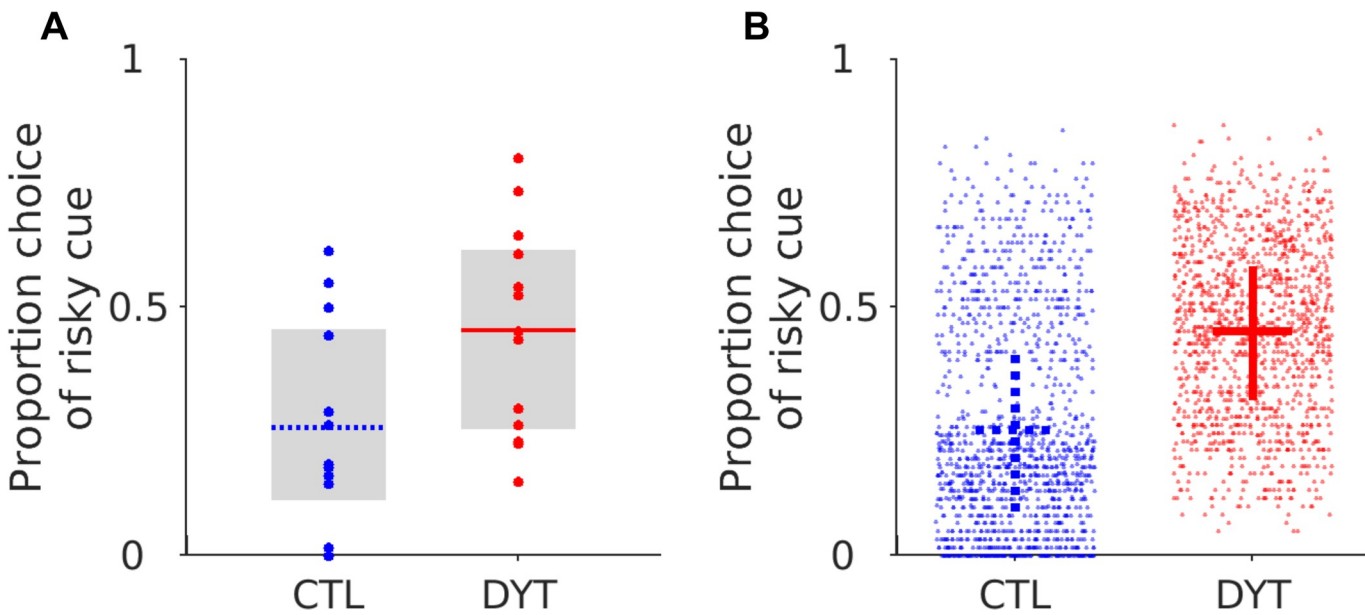

**Fig 4. Simulated choices under plasticity conditions for H2.** Example of a single simulated experiment using the final parameters estimates for the controls and patient estimates with H2 (**A**). This captures both the experimental mean and individual variance in both groups and closely replicates the experimental behaviour The CSP model was robust in replicating this behaviour across multiple simulations (**B**). For illustrative purposes we plot the first 100 of the 1000 simulated data sets from both the individual controls (blue) and patients (red). The mean choice of the risky cue and interquartile range (average between simulations) are represented by the dashed blue and solid red cross-hairs in the controls and patients respectively.

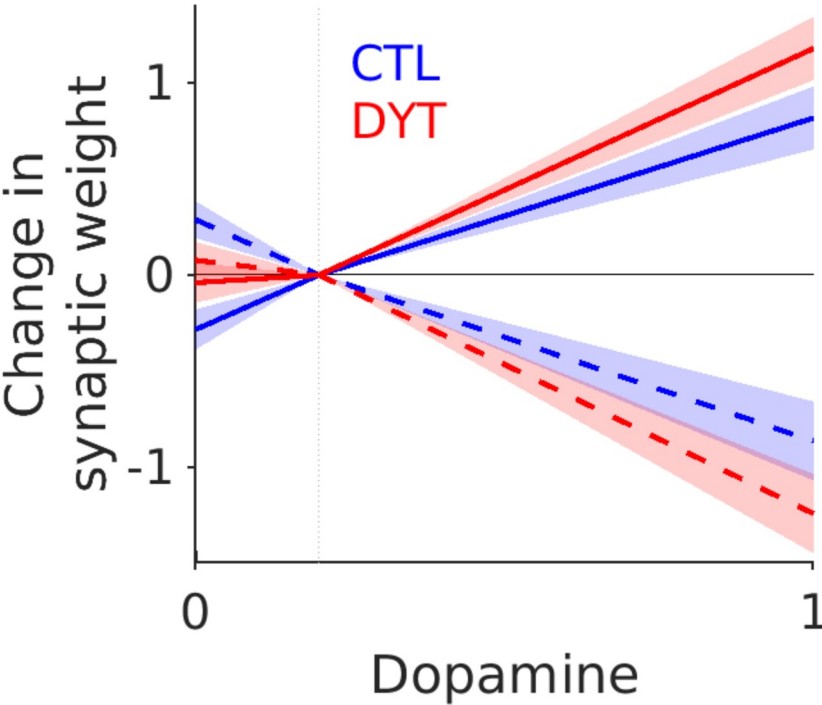

**Fig 5. Final dopamine-synaptic weight change for patients and controls.** Solid lines, D1R, dashed lines D2R. Mean values ± S.E.M represented by shaded area. Patients in red, controls in blue.

reduced below baseline levels following a loss, less LTP is produced at the cortico-striatal synapse in the models iMSNs and less LTD in the dMSN population. The behavioural consequences of these changes are for the model to choose the risky choice more frequently.

To understand why the CSP model could only recover the behaviour of the patients when cortico-striatal plasticity was abnormal in both groups of MSNs in opposite directions, we examined the time course of changes in D1 and D2R mediated cortico-striatal plasticity within the dMSN and iMSN populations in the model through the task. These are illustrated for H1 & H2 in Fig 5A and 5B. As expected for a striatum where the cortico-striatal synapse at dMSNs undergo greater LTP in response to a phasic increase in dopamine, the synaptic weight representing the risky cue in the patients increases rapidly to strengths that significantly exceed those of the controls in both models. In contrast, the cortico-striatal synaptic strength in the iMSNs, predominantly expressing the D2R, remains unchanged in the H2 model relative to the controls. At first glance, this seems counter intuitive given that H2 includes impaired iMSN cortico-striatal LTP (and increased LTD relative to LTP), however, this lack of build-up iMSN synaptic weight is pathological and reflects the blunted plasticity response to phasic reductions in dopamine that follow a risky "loss". This can be understood when the iMSN synaptic weight changes are compared between the H1 (Fig 6A) and H2 (Fig 6B) models. Under conditions of intact cortico-striatal plasticity at iMSNs, the H1 model generates a substantial increase in iMSN synaptic weight and activity in the indirect pathway. This is proportionate to the increased risky choices and the inevitable phasic reductions in dopamine that follow risky choices were the outcome is worse than expected. In contrast, in the presence of excessive LTD at cortico-striatal synapses in iMSNs under 'H2' plasticity conditions, there is no corresponding increase in indirect pathways weights. At a behavioural level this is indexed by no time dependent devaluation of the risky cue between blocks. This difference between the two

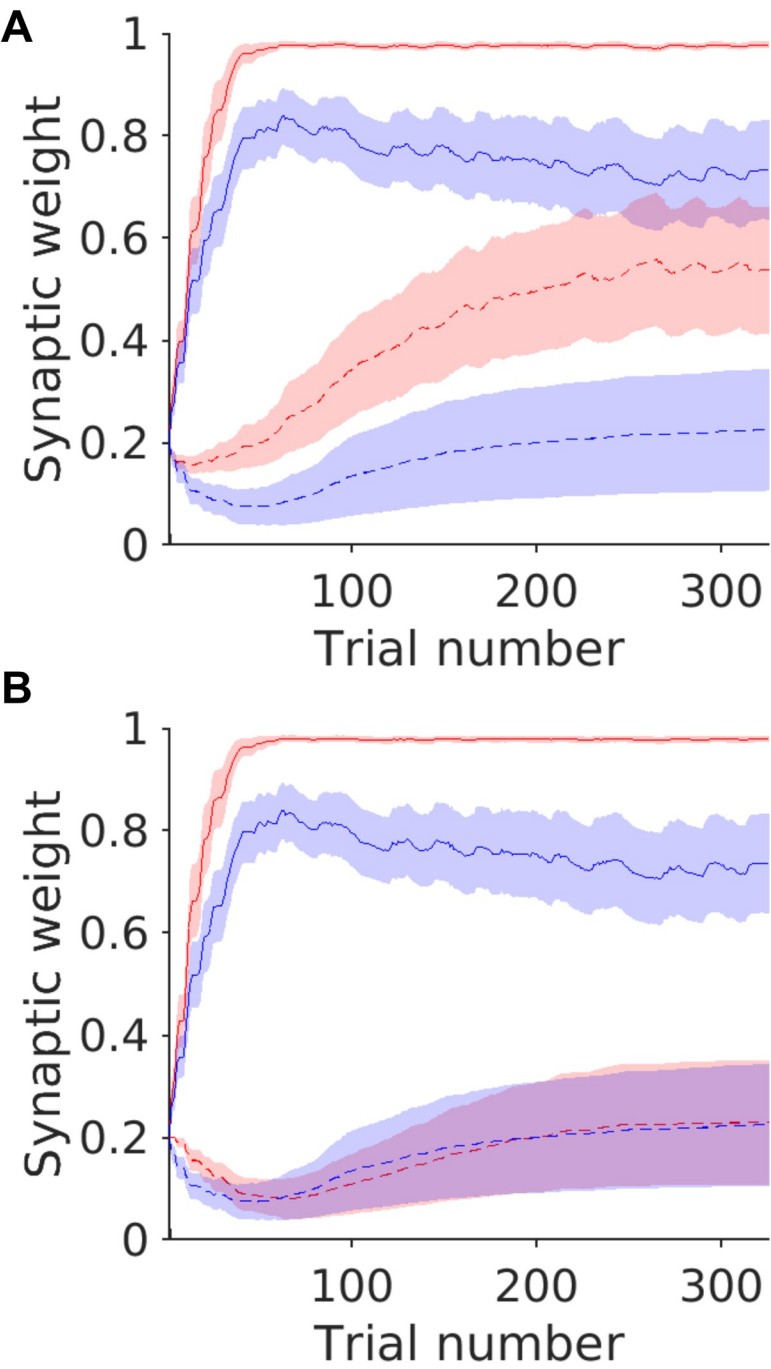

**Fig 6. Simulated striatal synaptic weight changes during the task for the risky cue.** Average simulated weights ± S.E. M (between simulations) for the CSP model under plasticity conditions of H1 (**A**) and H2 (**B**). Weights representing the risky cue are illustrated only. Cortico-striatal iMSN (D2R dependant) weights, dashed lines (—) dMSN (D1R dependant) cortico-striatal weights, solid lines (-). Patients in red, controls in blue.

models suggests that for the combination of reduced risk aversion and reduced choice devaluation observed in the DYT1 patients, cortico-striatal plasticity needs to be abnormal in opposing directions in both populations of dMSN and iMSNs of the direct and indirect pathways simultaneously.

## Discussion

Animal models for rare and devastating neurological diseases such as DYT1 dystonia are a central pillar to the development of the therapeutic armamentarium. Their relevance to the disease however requires close scrutiny and validation of experimental data across disciplines in order to maximise their translational potential. The purpose of the present study was aimed at reconciling recent experimental observations of risk neutral learning in patients with the TOR1A mutation [19] and reports of excess cortico-striatal LTP and diminished LTD in the established rodent genetic model of this disease [15]. To address this question, we used a computer model of the basal ganglia with detailed cortico-striatal plasticity which re-performed the same reinforcement learning task studied experimentally by Arkadir et. al (2016). This model was able to generate simulated choices that were statistically identical to those obtained experimentally in both the patient and control groups. Critical to the purpose of our study, was that the model was unable to reproduce the experimental behaviour of the patients when the pattern of plasticity abnormalities identified in the TOR1A rodents were incorporated into the model (our 'H3'–increased cortico-striatal LTP in both dMSN and iMSNs.). Only when the opposite pattern of cortico-striatal plasticity at iMSNs, ('H2' increased dMSN LTP and iMSN LTD), was the patients risk neutral choices replicated.

There are several limitations to our approach which necessitate caution when interpreting this result given that it conflicts the animal literature. Firstly, in order to capture the heuristic dynamics of synaptic plasticity at the cortico-striatal synapse, we reduced the mechanics of this process to a level of abstraction (four parameters) which makes direct comparison to *in vitro* measurements unclear. We cannot therefore make a meaningful quantitative comparison between our dopamine weight change curve (Fig 5) and the neurophysiological measurements from *in vitro* data. Reducing the biophysical detail of our model was necessary to avoid over fitting and allow meaningful hypothesis testing of the parameters of interest (in our case those relating to cortico-striatal plasticity). We cannot discount the possibility that assumptions of our model such as constant levels of tonic dopamine or omitting detailed cholinergic influence on iMSN cortico-striatal LTD [23] may have influenced our results. Future studies will be need to address whether these details are significant in determining the model's ability to replicate risky decision making.

Despite these limitations, there are several reasons to consider that the opposite iMSN plasticity abnormality to that seen in the animal data best explains the patient's behaviour. Firstly, the propensity to make risky choices can be considered a consequence of an enhanced sensitivity to reward following a successful risky choice, combined with blunted sensitivity to choices that lead to an aversive outcome. If the same increase in iMSN cortico-striatal LTP in the rodent model were manifest in the patients studied by Arkadir et. al., (2016), this would be expected to make them more risk averse by generating greater iMSN LTP following a risky choice which resulted in a losing outcome. In turn, this would raise the excitability of the indirect pathway and due to its net inhibitory influence on thalamo-cortical excitability, would suppress the likelihood of the risky cue being re-chosen. As optogenetic induction of cortico-striatal LTP in dMSNs leads to risk seeking [24] and iMSNs stimulation leads to risk aversion [25], the parallel increases in cortico-striatal LTP identified in both dMSN and iMSNs in TOR1A rodents would be expected to act antagonistically and nullify their overall effect on risk taking. This interpretation is supported by the results of our simulations as illustrated by the significant between block (Fig 3C 'H3') reduction in risky choices by the model with increased iMSN LTP, to risk taking levels comparable to that of the controls. This was neither a feature of the patient's experimental data or the simulated choices under model conditions of increased iMSN LTD ('H2'). Our simulations provide further evidence that increased LTP in dMSNs in combination with the opposite abnormality at iMSNs, of increased cortico-striatal LTD, is the most parsimonious explanation for the patient"s risk neutral behaviour.

Impaired generation of LTP at iMSN cortico-striatal synapses following a risky "loss" is in the DYT1 patients is also analogous to the loss of iMSN LTP and increased LTD in a model of impaired reversal learning seen in patients with cervical dystonia [21]. These results support a common mechanism of deficient LTP and excess LTD at iMSN cortico-striatal synapse's causing abnormal reinforcement learning that is independent of the specifics of the task or dystonia phenotype. As the density of D2R correlates with the sensitivity to negative decision outcomes [12, 13] the loss of cortico-striatal LTP at iMSN synapses is also consistent with imaging studies demonstrating reduced D2R in both forms of dystonia [26, 27].

The discrepancy between the human and rodent cortico-striatal iMSN plasticity abnormalities predicted by our simulations and those demonstrated in rodent models of DYT1 dystonia have crucial implications for our understanding of this condition and the development of new therapies for patients. In the first instance, they support the idea that striatal neurochemistry is not indifferent to dystonically manifest and non-manifest behavioural states. Notably, although animals with the TOR1A mutation have significant striatal neurochemical abnormalities, *they exhibit little to no phenotypic resemblance of a movement disorder*. It is conceivable therefore, that a reason for our results supporting an opposite pattern of abnormal plasticity at iMSNs to that seen *in vitro*, reflects a difference between the manifesting dystonic and non-manifesting states. This explanation is supported by observations from previous studies. First, following the peripheral nerve injury necessary to induce dystonia-like posturing in TOR1A mutant rodents, these are accompanied by significant increases in striatal dopamine and decreases in D2R receptor expression [28].This fundamental shift in dopaminergic neurochemistry has also been observed in recent post mortem studies comparing manifesting and non-manifesting carriers of the *TOR1A* mutant gene [29]. Second, the study of Edwards et. al., (2006) [30] emphasises the apparent paradox of how the same mutation can lead to an opposite physiological response depending on the clinically manifest state. Here TOR1A mutation carriers were tested using transcranial magnetic simulation protocols which induced LTD-like plasticity in healthy controls. These failed to induce any response in non-manifesting carriers but produced an exaggerated LTD-like response in the manifesting carriers.

Given this context it is unsurprising that our computational modelling of patient's behaviour converges on a conclusion opposite to that reported from experiments using animal models of human DYT1 dystonia. Our results have important implications for the development of small molecular therapies based on translational studies in rodents [31]. We argue that since performance on reinforcement learning tasks correlates with the severity of the movement disorder in these patients, these tasks could be used to screen putative therapeutic agents based upon their ability to modify reward learning. This would be a cost effective intermediate step prior to formal clinical trial testing aimed to at the identification of novel agents.

## Supporting information

**S1 Data. Experimental and simulated risk taking data under the different hypothesis tested.**
(MAT)

**S1 Raw data.**
(TXT)

## Acknowledgments

The authors thank Professor Mark Humphries for feedback on an earlier version of the manuscript. The authors would like to thank the editor and reviewers for their useful comments and suggestions, which greatly improved the manuscript.

## Author Contributions

**Conceptualization:** David Arkadir.

**Data curation:** David Arkadir.

**Formal analysis:** Tom Gilbertson.

**Methodology:** Tom Gilbertson.

**Software:** Tom Gilbertson.

**Writing – original draft:** Tom Gilbertson, David Arkadir, J. Douglas Steele.

**Writing – review & editing:** Tom Gilbertson, David Arkadir, J. Douglas Steele.

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
