## [Decision Letter · Decision Letter 0]

27 Jan 2020

PONE-D-19-33471

Opposing patterns of abnormal D1 and D2 receptor dependent cortico-striatal plasticity explain increased risk taking in patients with DYT1 dystonia

PLOS ONE

Dear Dr Gilbertson,

Thank you for submitting your manuscript to PLOS ONE. After careful consideration, we feel that it has merit but does not fully meet PLOS ONE’s publication criteria as it currently stands. Therefore, we invite you to submit a revised version of the manuscript that addresses the points raised during the review process.

Unfortunately I was only able to secure one review for this manuscript; however, I will act based on this one review as it's possible another review might lead to the same outcome, but with a longer delay.  As you can see the reviewer is quite critical of this work and argues for rejection.  After reading the manuscript myself I came very close to rejecting it, but I would like to give you the opportunity to revise the manuscript if you think you can address the reviewer's concerns.  If you choose to revise it then I will likely send it back to the original reviewer and also try to find a new reviewer.

We would appreciate receiving your revised manuscript by Mar 12 2020 11:59PM. To enhance the reproducibility of your results, we recommend that if applicable you deposit your laboratory protocols in protocols.io, where a protocol can be assigned its own identifier (DOI) such that it can be cited independently in the future. For instructions see: http://journals.plos.org/plosone/s/submission-guidelines#loc-laboratory-protocols

We look forward to receiving your revised manuscript.

Kind regards,

Darrell A. Worthy, Ph.D

Academic Editor

PLOS ONE

2. Please update your Methods section to indicate the name of the Institutional Review Board(s) that approved your study. Please also provide additional details regarding participant consent. In the Methods section, please ensure that you have specified (1) whether consent was informed and (2) what type you obtained (for instance, written or verbal). We appreciate that these details may have been provided in previous publications, but we would be grateful if you could please also include this information in your submission.

4. Please include your tables as part of your main manuscript and remove the individual files. Please note that supplementary tables (should remain/ be uploaded) as separate "supporting information" files.

Reviewers' comments:

Reviewer's Responses to Questions

**Comments to the Author**

1. Is the manuscript technically sound, and do the data support the conclusions?

Reviewer #1: Partly

2. Has the statistical analysis been performed appropriately and rigorously? 

Reviewer #1: Yes

3. Have the authors made all data underlying the findings in their manuscript fully available?

Reviewer #1: No

4. Is the manuscript presented in an intelligible fashion and written in standard English?

Reviewer #1: No

5. Review Comments to the Author

Reviewer #1: The research aims to reproduce a risk neutral behavior from human patients with mutated TOR1A gene. The study used a common reinforcement learning model and used weight (Wn) in updating action value (A) to represent synaptic strength. Furthermore, they made the change in synaptic strength (plasticity) influenced by dopamine (delta d). Then, they got the dopamine level for each trial and simulated the RPE. Finally, they adapt the competitive action selection model for the output of the D1- and D2-MSNs. The overall model looks good. However, the quality of the paper is severely harmed by several major issues.

1. The finding of the study doesn’t provide any insights. Although they claim their results on simulate D2-MSNs LTP is not in line with an in vitro data, this can also due to the limitation of the model not necessarily means additional factors existed. Note that the authors didn’t fully discuss the limitation of the model.

2. Authors seem cannot distinguish some basic ideas: cortico-striatal plasticity, synaptic strength, D1R- or D2R-expressing medium spiny neuron (or direct- and indirect-medium spiny neuron). For example, author wrote, "... post-synaptic abnormality in D2R transmission as the principle cause for ... ". There is no such receptor transmission. The idea might be dopamine transmission onto D2R-MSNs. Another example is "...D2R striato-pallidal LTP...". Authors should be aware that cortico-striatal LTP/LTD has nothing to do with the striato-pallidal and striato-nigral LTP. The cortico-striatal LTP/LTD does influence the output of D1- and D2-MSNs to GPe and SNr, but it doesn’t imply anything on striato-pallidal or -nigral LTP. Another relevant noticeable error is “… D1R-LTP…”. There is no such D1R-LTP. LTP cannot be induced on a receptor (let alone is a metabotropic receptor). I guess the authors want to express the idea: D1R-MSNs or D1-MSNs LTP. There are many more errors, which cause a huge misleading to non-field reader and research, like these above.

3. Citations are not appropriate. They mentioned the competition model but they didn’t cite Bariselli et al 2018 Brain Research paper, which proposed the competitive model to explain the action selection in the basal ganglia. Kravitzer et al., 2012 in the introduction cannot support RPE idea in the sentence. Frank 2005 in the introduction is also misleading. Frank did talk about how DA level affects plasticity. But, no direct statement about how positive and negative PRE shape or rely (actually are two different thing). The author also need to cite one more paper to support “rodent models” in the sentence “…proposed that patients with the TOR1A mutation should exhibit a learning strategy that is contingent with the abnormal plasticity seen in rodent models …’.

4. The late part of the manuscript has largely focused on the idea of RPE. But, the authors didn’t mention anything in the abstract. Actually, directly use DA level concept, but not RPE, will be more related to the cortico-striatal plasticity concept. Authors try to borrow a fancy idea like RPE but fail to test it. Although phasic dopamine level change correlates with the RPE, which has been shown in many studies, a direct way to do so is to do the regression on the behavior instead of just claim RPE because of dopamine level change. Taken together, using the concept of RPE doesn’t provide extra credit to paper.

6. PLOS authors have the option to publish the peer review history of their article (what does this mean?). If published, this will include your full peer review and any attached files.

Reviewer #1: No

---

## [Author Response · Author response to Decision Letter 0]

13 Feb 2020

Please see "response to reviewers" document uploaded with resubmission

---

## [Decision Letter · Decision Letter 1]

5 Mar 2020

PONE-D-19-33471R1

Opposing patterns of abnormal D1 and D2 receptor dependent cortico-striatal plasticity explain increased risk taking in patients with DYT1 dystonia

PLOS ONE

Dear Dr Gilbertson,

Thank you for submitting your manuscript to PLOS ONE. After careful consideration, we feel that it has merit but does not fully meet PLOS ONE’s publication criteria as it currently stands. Therefore, we invite you to submit a revised version of the manuscript that addresses the points raised during the review process.

I sent your manuscript back to the original reviewer.  The reviewer found the paper to be much improved, but noted a few things that need to be addressed before the paper is suitable for publication.  

We would appreciate receiving your revised manuscript by Apr 19 2020 11:59PM. To enhance the reproducibility of your results, we recommend that if applicable you deposit your laboratory protocols in protocols.io, where a protocol can be assigned its own identifier (DOI) such that it can be cited independently in the future. For instructions see: http://journals.plos.org/plosone/s/submission-guidelines#loc-laboratory-protocols

We look forward to receiving your revised manuscript.

Kind regards,

Darrell A. Worthy, Ph.D

Academic Editor

PLOS ONE

Reviewers' comments:

Reviewer's Responses to Questions

**Comments to the Author**

1. If the authors have adequately addressed your comments raised in a previous round of review and you feel that this manuscript is now acceptable for publication, you may indicate that here to bypass the “Comments to the Author” section, enter your conflict of interest statement in the “Confidential to Editor” section, and submit your "Accept" recommendation.

Reviewer #1: All comments have been addressed

2. Is the manuscript technically sound, and do the data support the conclusions?

Reviewer #1: Yes

3. Has the statistical analysis been performed appropriately and rigorously? 

Reviewer #1: Yes

4. Have the authors made all data underlying the findings in their manuscript fully available?

Reviewer #1: Yes

5. Is the manuscript presented in an intelligible fashion and written in standard English?

Reviewer #1: No

6. Review Comments to the Author

Reviewer #1: The revision significantly improve the manuscript. However, there are several things remain to be clarified.

1. Please try only use either direct-pathway/indirect-pathway MSN (dMSNs/iMSNs) or D1-/D2-MSNs (D1-/D2-MSNs) through out the manuscript. Because dMSNs can also express D2R and iMSNs can also express D1R. They are very similar idea but not equivalent.

2. Please revise the sentence, “Given the indirect pathways net inhibitory influence on cortical excitability ...”. I am not sure if this is some new findings that you found the GPi can back modulate cortical inputs in the striatum or this is pure wiring issue. I guess the information you try to delivery here is that the iMSNs play a negative role so that stimulation the corticostriatal activity onto iMSNs will suppresses the likelihood of a choice with an aversive outcome. So, please rephrase your sentence and make it more clear.

3. I think it also worthy to cite Ma et al., 2018, Nature Neuroscience paper. They are first few labs direct demonstrate how LTP and LTD on mPFC—> D1-MSNs and mPFC—> D2-MSNs change the behavior outcome and they also showed that of course block D1R and D2R affect the LTP and LTD induction on those two types of neurons.

7. PLOS authors have the option to publish the peer review history of their article (what does this mean?). If published, this will include your full peer review and any attached files.

Reviewer #1: No

---

## [Author Response · Author response to Decision Letter 1]

18 Mar 2020

1. Please try only use either direct-pathway/indirect-pathway MSN (dMSNs/iMSNs) or D1-/D2-MSNs (D1-/D2-MSNs) through out the manuscript. Because dMSNs can also express D2R and iMSNs can also express D1R. They are very similar idea but not equivalent.

In view of the dual expression of D1 and D2R on MSNs we have adopted a consistent nomenclature throughout the revised manuscript using “dMSN” and “iMSNs” exclusively. Accordingly we have removed reference to D1- or D2-MSNs to emphasise that the changes are in the direct and indirect pathways, rather than in the D1 or D1R subpopulations of MSNs given the co-expression of receptors. 

2. Please revise the sentence, “Given the indirect pathways net inhibitory influence on cortical excitability ...”. I am not sure if this is some new findings that you found the GPi can back modulate cortical inputs in the striatum or this is pure wiring issue. I guess the information you try to delivery here is that the iMSNs play a negative role so that stimulation the corticostriatal activity onto iMSNs will suppresses the likelihood of a choice with an aversive outcome. So, please rephrase your sentence and make it more clear.

This sentence in the introduction has been changed to:- “As the indirect pathway exerts an inhibitory influence on thalamo-cortical excitability, the effect of increased cortico-striatal synaptic strengthening at iMSNs is to suppress the likelihood of a choice with an aversive outcome being repeated”.

3. I think it also worthy to cite Ma et al., 2018, Nature Neuroscience paper. They are first few labs direct demonstrate how LTP and LTD on mPFC—> D1-MSNs and mPFC—> D2-MSNs change the behavior outcome and they also showed that of course block D1R and D2R affect the LTP and LTD induction on those two types of neurons.

Ma et al [24] now cited in the discussion:- “As optogenetic induction of cortico-striatal LTP in dMSNs leads to risk seeking [24] and iMSNs stimulation leads to risk aversion [25], the parallel increases in cortico-striatal LTP identified in both dMSN and iMSNs in TOR1A rodents would be expected to act antagonistically and nullify their overall effect on risk taking.

---

## [Editor Report · Decision Letter 2]

6 Apr 2020

Opposing patterns of abnormal D1 and D2 receptor dependent cortico-striatal plasticity explain increased risk taking in patients with DYT1 dystonia

PONE-D-19-33471R2

Dear Dr. Gilbertson,

We are pleased to inform you that your manuscript has been judged scientifically suitable for publication and will be formally accepted for publication once it complies with all outstanding technical requirements.

With kind regards,

Darrell A. Worthy, Ph.D

Academic Editor

PLOS ONE
---

## [Editor Report · Acceptance letter]

22 Apr 2020

PONE-D-19-33471R2 

Opposing patterns of abnormal D1 and D2 receptor dependent cortico-striatal plasticity explain increased risk taking in patients with DYT1 dystonia 

Dear Dr. Gilbertson:

I am pleased to inform you that your manuscript has been deemed suitable for publication in PLOS ONE. Congratulations! Your manuscript is now with our production department. 

With kind regards,

on behalf of

Dr. Darrell A. Worthy 

Academic Editor

PLOS ONE